# Geospatial Metabolomics Unravel Regional Disparities in Sedative Compounds and Volatile Profiles of *Ziziphi Spinosae Semen* Across Chinese Production Areas

**DOI:** 10.3390/plants14172739

**Published:** 2025-09-02

**Authors:** Jia Tian, Shujuan Hou, Hanbing Zhu, Ruirui Dao, Junguang Ning, Peixing Ren, Fuxu Pan, Mengjun Liu, Zhihui Zhao

**Affiliations:** 1College of Horticulture, Hebei Agricultural University, Baoding 071001, China; 15612276351@163.com (J.T.); hsj15131208376@126.com (S.H.); 15719648726@163.com (R.D.); 15530766977@163.com (J.N.); rpx980529abc@163.com (P.R.); pfx1234562022@163.com (F.P.); 2College of Food Science and Technology, Hebei Agricultural University, Baoding 071001, China; zhuhb77@163.com; 3Research Center of Chinese Jujube, Hebei Agricultural University, Baoding 071001, China

**Keywords:** *Ziziphi Spinosae Semen*, metabolites, sleeping ingredients, metabolomics, volatile compounds

## Abstract

*Ziziphi Spinosae Semen* (ZSS) has significant medicinal value, and its growing environment critically influences medicinal component accumulation. We analyzed 10 ZSS samples from six major Chinese production areas, identifying 2994 metabolites while exploring tranquilizing constituents and volatiles. Lipids and amino acids were the primary nutrients, while terpenoids were the most abundant class of secondary metabolites. Volatile profiling revealed characteristic sour-fruity-herbaceous flavors, with GS-QY samples showing the highest volatile content. HB-XT and LN-CY samples accumulated the most sedative compounds (jujubosides A/B, spinosin). These findings demonstrate production regions significantly influence ZSS’s medicinal/aromatic profiles, supporting targeted product development.

## 1. Introduction

Sour jujube, also referred to as wild jujube or acid jujube (*Ziziphus acidojujuba* Cheng et Liu–Z. *jujuba* Mill. var. *spinosa* Hu), is a deciduous plant that belongs to the Rhamnaceous family and is characterized by thorny branches. This species is not only the wild ancestor of the cultivated jujube fruit tree, but also an important fruit tree cultivated in China [1]. *Ziziphi Spinosae Semen* (ZSS) is the dried seed of the wild jujube, known as suan zao ren in Chinese. ZSS is a widely used herb in traditional Chinese medicine, known for its ability to soothe the mind and alleviate insomnia [2]. Its historical use dates to Shennong Bencao Jing (ca. 200 CE), having been utilized in traditional Asian medicine for millennia to address mild anxiety, nervous agitation, and sleep disorders. ZSS contains a complex mixture of pharmacological bioactive compounds, with the main secondary metabolites identified through analytical techniques being cyclopeptide alkaloids, triterpene saponins, triterpene acids, flavone C-glycosides, and unsaturated fatty acids [3]. These metabolites have demonstrated various beneficial effects, including enhancing cognitive function through anticholinesterase activity [4], protecting against cell damage induced by NMDA [5], and improving immunological activity [6]. The primary tranquilizing ingredients found in ZSS are Jujubosides and Spinosin. A high dose of jujuboside has been found to effectively suppress the hyperactivity of the hippocampal CA1 area induced by penicillin sodium [7,8], while spinosin significantly enhances pentobarbital-induced sleep in rats [9]. Flavonoids in the aqueous extract of ZSS may also play a role in the treatment of insomnia [10,11].

China boasts abundant wild jujube germplasm resources, mainly grown in provinces such as Hebei, Shandong, Shanxi, Shaanxi, Liaoning, and Gansu. Evaluating the nutrient content and quality of ZSS from different habitats plays a critical role in the development of functional foods and potential pharmaceuticals. However, comprehensive evaluations of ZSS quality across various regions remain limited. This study employed a widely targeted metabolomics approach to analyze metabolites in 10 ZSS samples collected from six major production areas in China. Additionally, HPLC was used to quantify jujuboside A, jujuboside B, and spinosin and to verify the results from LC-MS, and the compositional characteristics of volatile compounds were analyzed using GC-IMS.

Our comprehensive evaluation is based on the ranking of primary metabolites and the total content of key components related to sleep aids. The goal is to gain a deeper understanding of the distribution of these essential constituents across regions and to explore the composition of volatile components in ZSS. These findings will help facilitate the widespread processing and utilization of ZSS, advancing its applications in both medicinal and culinary fields.

## 2. Results

### 2.1. Metabolite Profiling and Classification of ZSS

Liquid chromatography-mass spectrometry (LC-MS) analysis identified 2994 metabolites across 10 ZSS samples from major production regions. These metabolites were classified into 22 categories (Figure 1a), with the 6 most abundant classes being lipids and lipid-like molecules (569 compounds), amino acids, peptides, and analogues (445), organic acids and derivatives (310), terpenoids (303), organoheterocyclic compounds (288), carbohydrates, and carbohydrate conjugates (169).

Cluster analysis of bioactive compounds revealed distinct regional patterns (Figure 1b). The sample from GS-QY (Gansu) exhibited the highest overall abundance of saponins, flavonoids, alkaloids, and fatty acids. The sample from HB-ZH (Hebei) showed the highest flavonoid content (*p* < 0.05), suggesting potential for flavonoid-based therapeutics. The sample from LN-CY (Liaoning) accumulated significantly higher levels of saponins and alkaloids but lower flavonoids, indicating region-specific metabolic specialization, while Shaanxi-derived ZSS (Sx-WN) was enriched with carbohydrates (*p* < 0.01), highlighting their utility in energy-supplement applications.

### 2.2. Geographic Variation in Metabolic Profiles

Principal component analysis (PCA) explained 72.30% of total variance (PC1 + PC2), demonstrating clear separation among regional samples (Figure 2a). Quality control (QC) samples clustered tightly, confirming analytical reproducibility.

Correlation analysis (|r| ≥ 0.5) revealed strong intra-regional correlations (e.g., SX-YC, SX-CZ, SX-LL; r > 0.8) and significant cross-regional associations; significant cross-regional associations between LN-CY (Liaoning) and HB-CD (Hebei) (*r* = 0.54) were attributable to their shared location in the Yanshan Mountains (Figure 2b). In contrast, samples from the Taihang Mountains (HB-ZH, HB-XT) formed a distinct cluster. Hierarchical clustering analysis showed samples segregated into three geographic clusters (Figure 2c): the Taihang group, with six samples from Hebei, Shanxi, and Shandong (HB-ZH, HB-XT, SX-CZ, SX-YC, SD-LY, SX-LL); the Northwestern group, with two samples (Gansu, Shaanxi); and the Yanshan group, with HB-CD (Hebei) and LN-CY (Liaoning).

Comparative metabolomic profiling revealed significant regional variations in metabolite composition. The LN-CY samples (Chaoyang, Liaoning Province) exhibited the most distinct metabolic profile, with notably higher levels of 20 characteristic compounds. These included armillatin, a bioactive flavonoid with demonstrated antimicrobial properties, and alaproclate, a structural analog of serotonin reuptake inhibitors. Additionally, this region showed prominent accumulation of benzoic acid and phenol derivatives, collectively contributing to its enhanced preservative and antimicrobial potential [12]. In contrast, HB-CD samples (Hebei province) displayed 11 differentially abundant metabolites, including isoleucyl-alanine, pseudooxynicotine, and dethiobiotin. Most notably the pharmaceutical excipient tromethamine and the sulfur-containing amino acid derivative L-2-amino-5-(methylthio) pentanoic acid. Meanwhile, GS-QY samples (Gansu province) demonstrated a unique metabolic pattern characterized by 10 region-specific compounds, including the prostaglandin precursor 11-deoxy PGF2α and the modified sterol 25-azacholesterol. These region-specific metabolic patterns may arise from three key factors: differential biosynthetic pathway activity, environmental modulation of secondary metabolism, and genetic divergence among geographically distinct ZSS populations (region-specific metabolites detailed in Appendix A).

### 2.3. Quantification of Bioactive Compounds

Quantitative HPLC analysis of key sedative components revealed significant regional variations in ZSS phytochemical composition. Spinosin emerged as the most abundant bioactive compound across all samples (0.081–0.124%), with notably higher concentrations observed in LN-CY and SD-LY accessions (Figure 3c). The content of jujuboside A exhibited a broad range from 0.032% to 0.068%, with GS-QY samples containing significantly elevated levels compared to other regions (*p* < 0.01). Notably, jujuboside B showed remarkable regional specificity, being predominantly detected in LN-CY samples at 0.054%, a concentration that significantly surpassed all other accessions (*p* < 0.001).

The cumulative content analysis of these three major sedative compounds identified LN-CY, HB-CD, and HB-XT as possessing the highest combined concentrations (Figure 3d), suggesting these regional variants may offer superior sleep-enhancing potential. Complementary quantification of secondary metabolites demonstrated that total saponins averaged 0.75% across all samples, reaching peak levels of 1.12% in GS-QY. In contrast, total flavonoid content averaged 1.59%, with HB-ZH samples containing 2.3-fold higher concentrations than LN-CY accessions, highlighting substantial chemotypic variation among production regions.

### 2.4. Metabolic Pathway Enrichment

The KEGG pathway enrichment analysis revealed that 375 identified metabolites in ZSS were involved in 95 distinct metabolic pathways, with the most significantly enriched pathways (*p* < 0.001) primarily related to amino acid metabolism and secondary metabolite biosynthesis. Notably, tryptophan metabolism was the top enriched pathway, suggesting its crucial role in serotonin and melatonin synthesis that may contribute to ZSS’s neuropharmacological effects [13]. The tryptophan biosynthetic pathway plays a critical role in coordinating plant growth and stress responses [14]. As the wild jujube tree primarily grows in the mountainous regions of northern China, adverse conditions such as low temperatures and significant diurnal temperature variations may promote tryptophan accumulation, thereby enhancing the synthesis of downstream neuroactive metabolites [15]. The significant enrichment of ABC transporters (EIP) indicated active transmembrane transport of bioactive compounds [16], while phenylpropanoid biosynthesis reflected ZSS’s abundant flavonoid production associated with its antioxidant properties [17]. Furthermore, the marked enrichment of terpenoid and alkaloid biosynthesis pathways aligned well with ZSS’s documented pharmacological activities, particularly its sedative and neuroprotective effects [18]. These findings systematically elucidate the metabolic basis of ZSS’s therapeutic properties, highlighting its regulatory roles in neurotransmitter metabolism, stress response, and production of pharmacologically active compounds (Figure 4).

### 2.5. Analysis of Volatile Organic Compounds in ZSS

We used GC-IMS to obtain comprehensive ion mobility spectrum, construct three-dimensional topographic maps, and analyze parameters such as residence time (seconds), relative ion migration time (arbitrary units), and peak intensity (volts). The three-dimensional topographic maps (Figure 5a) reveal varying peak intensities across ZSS samples from different regions, The analysis reveals that, while the types of volatile compounds are similar across samples from different regions, the primary differences lie in their concentration levels.

Additionally, a two-dimensional, top-down view of the ZSS VOCs was created (Figure 5b). The vertical axis shows GC retention time, while the horizontal axis represents ion migration time (during normalization). The red line at 1.0 marks the reactive ion peak (RIP after normalization), with differently shaded and sized points on either side indicating VOC detection results. The whiter the dots, the lower the detected volatile content; redder dots correspond to higher volatile content. It can be observed that most signals appear within the retention time range of 200 to 800 s and the drift time range of 1.0 to 1.6 s. To gain a deeper insight into the differences, a comparison mode was applied. Using the spectra of the SX-LL sample as a reference, the spectra from other samples were analyzed to generate a difference map for comparison across regions (Figure 5c). In this map, if the VOC content of a sample is equal to the reference; the background appears white, while red signifies higher content, and blue represents lower content. The results demonstrate that the samples from GS-QY, SX-YC, SX-WN, and HB-ZH exhibit relatively similar compositional profiles.

Through qualitative analysis of the volatile compounds, a total of 51 peaks were observed. The results of the library search qualitative analysis for the samples are shown in Figure 5d. In total, 51 signal peaks were detected, from which 34 volatile compounds were identified. These include 7 alcohols, 7 esters, 7 ketones, 4 olefins, 4 aldehydes, 5 other substances, and 3 unidentified compounds (Appendix A). Some compounds like 3-Methylbutan-1-ol, Butanol, and 12 other compounds can generate multiple signals at varying concentrations, such as in the form of monomers, dimers, or trimers. We recorded it once.

As shown in Figure 5e, the GS-QY sample exhibited distinctive flavor characteristics, with significantly higher concentrations of hexanal, isoamyl formate, 2-pentanone, ethyl propionate, and ethyl 2-methylpropanoate compared to other regions. The SX-YC sample contained the highest level of α-pinene. Samples from adjacent Taihang Mountain regions (SX-CZ, HB-XT, HB-ZH) showed similar aldehyde profiles, with HB-ZH displaying the highest concentrations of 2,3-pentanedione, 3-methylbutan-1-ol, 3-methyl-2-butanol, (E)-2-heptenal, and β-pinene. Notably, HB-XT was characterized by elevated levels of β-myrcene. Meanwhile, LN-CY demonstrated the highest propanol content, indicating significant regional differentiation.

To accurately identify the significantly altered volatile compounds, OPLS-DA was employed to differentiate observation groups and identify key variables responsible for intergroup differences. Figure 6a demonstrates excellent separation between different samples. The OPLS-DA model exhibited strong explanatory power with fitting parameters (R^2^X = 0.992, R^2^Y = 0.984, Q^2^ = 0.954). Model robustness was verified through 200 permutation tests, where the Q^2^ regression line below zero (R^2^ = 0.275, Q^2^ = −0.865) in Figure 6b confirmed reliability without overfitting. GC-IMS analysis identified 10 compounds with VIP > 1 (Figure 6c), all showing significant intergroup differences (*p* < 0.05) as potential characteristic markers. These compounds were categorized into three flavor classes: organic acids (e.g., acetic acid) imparting sour notes, esters (e.g., ethyl propionate) contributing fruity tones, and carbonyls (e.g., 3-hydroxy-2-butanone) providing creamy and grassy flavors. High-VIP compounds (VIP > 2.0) like acetic acid and ethyl propionate served as key flavor markers, offering crucial insights into sample flavor characteristics. These findings not only reveal the diversity of volatile components but also provide theoretical support for flavor regulation.

## 3. Discussion

*Ziziphi Spinosae Semen* (ZSS), a traditional Chinese medicinal material, has demonstrated significant pharmacological activities including anti-anxiety [19], anti-depression [20], neuroprotection [21], cardiotonic effects [22], and sleep improvement [23]. This study elucidated the intrinsic relationship between chemical composition and geographical origin of *Ziziphi Spinosae Semen* through metabolomics analysis, thereby establishing a theoretical foundation for the in-depth development of related products.

Our integrated metabolomics approach identified 2994 metabolites, revealing distinct chemotypic variations among ZSS samples from different production regions. Notably, lipid compounds constituted the most abundant metabolite class (19% of total metabolites), consistent with the characteristic oil-rich composition of this seed-based medicinal material [24]. This finding supports the traditional use of ZSS as a nutrient-rich therapeutic agent.

Our findings revealed that the LN-CY (Chaoyang Liaoning province), HB-CD (Chengde Hebei province), and HB-XT (Xingtai Hebei province) production areas exhibited the highest combined concentrations of three major sedative compounds (jujuboside A, jujuboside B, and spinosin). Notably, the ZSS from Xingtai, Hebei Province (commonly known as “Xing Zao Ren”), is characterized by plump, red, and glossy seeds with a long cultivation history, which can be attributed to the region’s unique sandy alkaline soil and arid climate. Furthermore, the Chaoyang region in Liaoning Province has emerged as a promising high-quality production area worthy of attention. These results suggest that materials from these regions may possess enhanced sleep-promoting properties.

Observation was further supported by KEGG pathway analysis, which identified tryptophan metabolism as a potentially crucial pathway, with tryptophan serving as a metabolic precursor for serotonin and melatonin synthesis [25]. As an endogenous sleep-regulating substance in humans, melatonin exhibits multiple physiological functions, including circadian rhythm regulation, sleep latency reduction, and sleep quality improvement [26]. These findings provide novel scientific clues for understanding the pharmacological mechanisms underlying ZSS’s sedative-hypnotic effects.

Principal component analysis (PCA) of the metabolic profiles yielded particularly noteworthy findings regarding geographical influences. The clustering of samples from Chengde, Hebei (HB-CD), with those from Chaoyang, Liaoning (LN-CY), rather than with samples from other Hebei production areas (Zanhuang and Xingtai), clearly demonstrates that administrative boundaries do not determine metabolic profiles. Instead, our data strongly suggest that climatic conditions represent the primary environmental factor governing metabolite accumulation patterns in ZSS. This conclusion has important implications for the cultivation and quality control of this medicinal material.

The main volatile components in *Ziziphi Spinosae Semen* (ZSS) collectively form its characteristic flavor profile, consisting of organic acids (acetic acid, etc.), esters (ethyl acetate, ethyl propionate, etc.), alcohols (3-methyl-1-butanol, etc.), and carbonyl compounds (2,3-pentanedione, etc.). These components synergistically create the distinctive flavor characteristics of ZSS, manifested as prominent nutty, sweet-wine, and complex fruity notes, complemented by fresh grassy and pine-like aromas [27,28,29,30,31]. Significant flavor variations were observed among samples from different regions. The GS-QY sample exhibits intense fruity and grassy notes due to high concentrations of hexanal, isoamyl formate, and ethyl propionate [27,29]. Meanwhile, the HB-ZH sample displays a pronounced creamy-nutty character, owing to elevated levels of 2,3-pentanedione and (E)-2-heptenal [29]. The SX-YC sample demonstrates a distinct fresh pinewood aroma from α-pinene [32], while the HB-XT sample features a signature citrus note imparted by β-myrcene [33], potentially yielding significant effects in aromatherapy applications [34]. This not only reveals regional specificity but also provides an important basis for developing distinctive aromatic products.

Although the samples covered 10 production regions across six provinces (with 30 biological replicates in total), meeting the basic sample size requirements for omics studies, certain provinces (e.g., GS-QY, SD-LY, and LN-CY) contained only a single sampling site, which limited comprehensive characterization of intra-province chemical diversity. Furthermore, the effects of spatial heterogeneity of wild resources and interannual climate fluctuations on metabolite profiles could not be fully captured due to the single-year sampling design. Future studies should expand the sampling to include more sites across broader geographical ranges, incorporate multi-year dynamic sampling, and integrate genetic analytical approaches such as DNA barcoding or population genomics to systematically clarify the interactive effects of genetic and environmental factors on the metabolic diversity of ZSS.

## 4. Materials and Methods

### 4.1. Experimental Materials

*Ziziphi Spinosae Semen* (ZSS) samples were collected from wild jujube (*Ziziphus acidojujuba* Cheng et Liu–Z. *jujuba* Mill. var. *spinosa* Hu) plants across 10 geographical locations in 6 major production provinces in China during the peak maturity period (September to October 2022) (Figure 7). Detailed geographical coordinates and climatic characteristics of each sampling location are provided in Appendix A.

At each location, three biological replicates were collected within a proximate area. Each replicate consisted of seeds pooled from 10 individual trees, yielding a total of 30 independent samples (10 locations × 3 replicates). Mature fruits were harvested, and seeds were carefully extracted after removal of the pulp and cracking of the stones. All samples were uniformly dried at 35 °C for 48 h, sealed in airtight bags, and stored at −20 °C until further analysis to preserve metabolic integrity.

ZSS was identified as the dry mature seed of Ziziphus acidojujuba Cheng et Liu–Z. *jujuba* Mill. var. *spinosa* Hu by Mengjun Liu (Professor of Hebei Agricultural University).

Liquid chromatography grade solvents methanol and acetonitrile were purchased from Fisher Chemical. Jujuboside A, jujuboside B, Spinosin, and rutin were purchased from Shanghaiyuanye Co., Ltd. (Shanghai, China).

### 4.2. Sample Preparation

#### 4.2.1. Sample Preparation for Metabolomics Analysis

The dried ZSS were ground, and 50 mg of the powder was weighed into a 2 mL centrifuge tube. Next, 400 μL of methanol solution (methanol: water = 4:1 (*v*/*v*)) was added, and the samples were ground with a grinder at −10 °C for 6 min. They were then extracted with a low-temperature ultrasonic wave of 40 KHz at 5 °C for 30 min, and then the sample was allowed to stand still at −20 °C for 30 min. They were then centrifuged for 15 min (13,000 g, 4 °C), and the supernatant was transferred to the injection vial with internal intubation for the analysis. Additionally, 20 μL of supernatant from each sample was pooled as a quality control sample. Each sample was analyzed in triplicate, with each result representing the average of three technical replicates.

#### 4.2.2. Sample Preparation for Total Flavonoids, Saponins, Jujuboside A, Jujuboside B, and Spinosin Analysis

To prepare the sample, 1.5 g of degreased ZSS powder was subjected to ultrasound extraction using 30 mL of 95% ethanol for 50 min. Then, the mixture was allowed to stand for 30 min and centrifuged at 9400 rpm. The supernatant was then collected, with one portion used for the determination of total flavonoids and saponins, while another portion was concentrated to dryness using a rotary evaporator. The dried residue was then redissolved in 3 mL of methanol in a centrifuge tube and filtered through a 0.22 μm membrane filter to ensure purity and remove particulate matter for subsequent analysis of jujuboside A, jujuboside B, and spinosin.

### 4.3. LC-MS Analysis Method

The samples were separated on a BEH C18 column (100 mm × 2.1 mm i.d., 1.7 µm) prior to mass spectrometric detection. Mobile phase A consisted of 2% acetonitrile in water (containing 0.1% formic acid), while mobile phase B was acetonitrile (containing 0.1% formic acid). The gradient elution program was as follows: initial 2% B (0–0.5 min), increased to 35% B (0.5–7.5 min), then ramped to 95% B (7.5–13 min), held at 95% B for 1.4 min (13–14.4 min), rapidly returned to 2% B (14.4–14.5 min), and finally maintained at 2% B (14.5–16 min). The flow rate was maintained at 0.40 mL/min with a column temperature of 40 °C. Mass spectrometric data acquisition was performed in both positive and negative ion modes with a mass scan range of m/z 70–1050. The ion source parameters were set as follows: spray voltage +3500 V (positive mode) and −3000 V (negative mode), sheath gas flow 50 psi, auxiliary gas flow 13 psi, and ion transfer tube temperature 450 °C. Data-dependent MS/MS analysis was conducted using stepped normalized collision energies of 20, 40, and 60 eV. The mass resolution was set to 70,000 for full MS scans and 17,500 for MS/MS scans.

### 4.4. GC-IMS Analysis Method

Volatile compounds from ZSS were analyzed with GC-IMS using a FlavourSpec^®^ instrument (G.A.S., Dortmund, Germany); 1 g of dry ZSS powder was weighed and placed in a 20 mL headspace glass extraction flask. The sample was incubated at 60 °C for 15 min. Following incubation, a 0.5 mL headspace phase solution was automatically injected into the injector using a syringe with a 45 °C splitless injection. Volatile components were separated by gas chromatography using a capillary column MXT-WAX (30 m × 0.53 mmID, 1.0 µm df, RESTEK, Corporation, Bellefonte, PA, USA) and later coupled in 4 IMS. The carrier gas was nitrogen (99.999% purity). The programmed flow rate was 2 mL/min up to 2 min, 10 mL/min up to 8 min, and terminated at 15 min when the flow rate was increased to 100 mL/min. The test was carried out under atmospheric pressure. The substances to be tested were first separated at 60 °C and then ionized at 45 °C in an ion mobility chamber. The specimens were divided into three groups for determination. C4-C8 ortho-ketone (China Pharmaceutical and Chemical Reagents Beijing Co., Ltd., Beijing, China) was used as an external reference, and the retention index (RI) of each component was calculated under the same chromatographic conditions as the specimens. Ultimately, the identification of volatiles was achieved by comparison with the standard drift time (the time required for ion drift).

### 4.5. HPLC-UV Analysis Method

We conducted HPLC-UV analysis using Agilent 1200 series instruments. The analysis included jujuboside A, jujuboside B, and spinosin. We used a Diamonsil C18 column (5 μm, 250 × 4.6 mm), and, for the analysis of jujuboside A and jujuboside B, we used a detection wavelength of 204 nm. The mobile phase consisted of acetonitrile and ultrapure water with a ratio of 35:65. The column temperature was maintained at 30 °C. The flow rate was set to 0.8 mL/min, and the injection volume was 10 μL.

For the detection of spinosin, we used a detection wavelength of 335 nm. The mobile phase for this analysis consisted of acetonitrile and ultrapure water in a ratio of 20:80. The column temperature was maintained at 25 °C. The flow rate was set to 0.8 mL/min, and the injection volume was 10 μL.

The preparation of the reference solution, with 1 mg each of jujuboside A, jujuboside B, and spinosin, accurately weighed, was added into 5 mL volumetric flasks, respectively, with the volume fixed to the scale line with methanol, prepared with 0.2 mg/mL of standard solution, diluted into solutions of different concentrations for standby, and injected in triplicates under the chromatographic conditions mentioned before.

### 4.6. UV-5200PC Spectrophotometer Analysis Method

We used UV-5200PC spectrophotometer from Shanghai Yiheng Instrument Co., Ltd. (Shanghai, China) to quantify the total saponins and flavonoids in ZSS with jujuboside B and ruitn as references, respectively, and 2 mg of jujuboside B standard and rutin standard were individually weighed and transferred into 5 mL brown volumetric flasks. Each standard was dissolved in methanol, and the resulting solutions were further diluted to create reference solutions with varying proportions.

### 4.7. Data Analysis

Principal component analysis (PCA), partial least squares-discriminant analysis (PLS-DA), and correlation analysis were performed using the Majorbio Cloud Platform (https://www.majorbio.com (accessed on 1 January 2023)). Metabolic pathway enrichment analysis was conducted based on the KEGG database. Data of nutritional components and bioactive constituents were analyzed using OriginPro 2021 (Version 9.8.0.200; OriginLab Corporation, Northampton, MA, USA) under default parameters. Excel 2016 (Version 16.0.4266.1001; Microsoft Corporation, Redmond, WA, USA) was employed for data organization and preliminary processing of the content of major medicinal components. A one-way analysis of variance (ANOVA) was carried out using SPSS 26.0 (Version 26.0.0.0; SPSS Inc., Chicago, IL, USA), with significant differences among groups further compared using Duncan’s test (*p* < 0.05). Additionally, an orthogonal partial least squares-discriminant analysis (OPLS-DA) model was established using SIMCA 14.1 (Version 14.1.0.2047; Umetrics, Malmö, Sweden) to perform supervised multivariate analysis for assessing relationships between sample groups based on their metabolic similarities and differences.

## 5. Conclusions

This study systematically elucidated the chemical–geographical relationships of Ziziphi Spinosae Semen (ZSS) through integrated metabolomics and volatile component profiling. Analytical results demonstrated that samples from Xingtai (Hebei) and Chaoyang (Liaoning) exhibited superior profiles of sedative-hypnotic compounds. KEGG pathway analysis revealed that tryptophan metabolism likely serves as the crucial pathway underlying its sleep-enhancing effects. Principal component analysis indicated that climatic conditions, rather than administrative boundaries, predominantly determined the metabolic patterns. Furthermore, characteristic flavor profiles were identified across different regions, notably intense fruity and grassy notes in Qingyang (Gansu) samples and distinct creamy-nutty aromas in Zanhuang (Hebei) samples offering valuable references for specialty product development.

## Figures and Tables

**Figure 1 plants-14-02739-f001:**
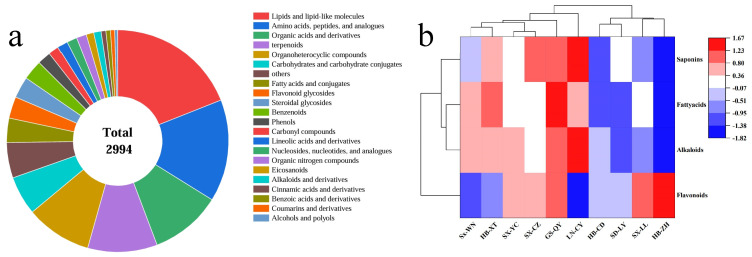
Cluster analysis of nutritional and bioactive components in *Ziziphi Spinosae Semen* samples from different main production areas; (**a**) cluster analysis of all nutritional components; (**b**) cluster analysis of saponins, flavonoids, alkaloids, and fatty acids.

**Figure 2 plants-14-02739-f002:**
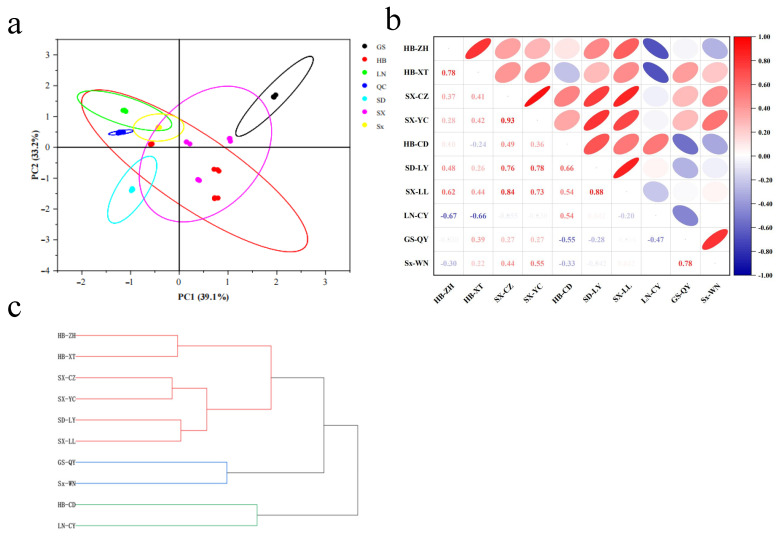
Correlation analysis of *Ziziphi Spinosae Semen* samples from different main production areas. (**a**) PCA of *Ziziphi Spinosae Semen*. (**b**) Intragroup correlation analysis of *Ziziphi Spinosae Semen* samples. (**c**) Cluster analysis of samples.

**Figure 3 plants-14-02739-f003:**
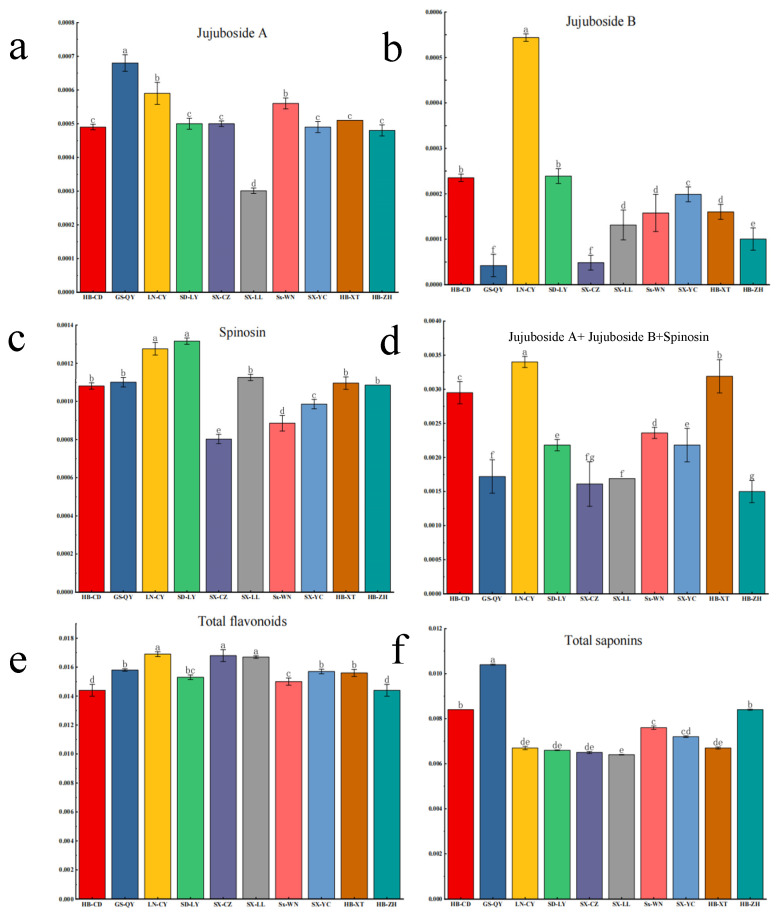
Analysis of the content of main medicinal components in *Ziziphi Spinosae Semen*. (**a**) content of Jujuboside A; (**b**) content of Jujuboside B; (**c**) content of Spinosin; (**d**) Cumulative Content of Jujuboside A, Jujuboside B and Spinosin; (**e**) content of total flavonoids; (**f**) content of total saponins. Values represent the average, and error bars represent standard errors. Different letters above the columns indicate significant differences (*p* < 0.05).

**Figure 4 plants-14-02739-f004:**
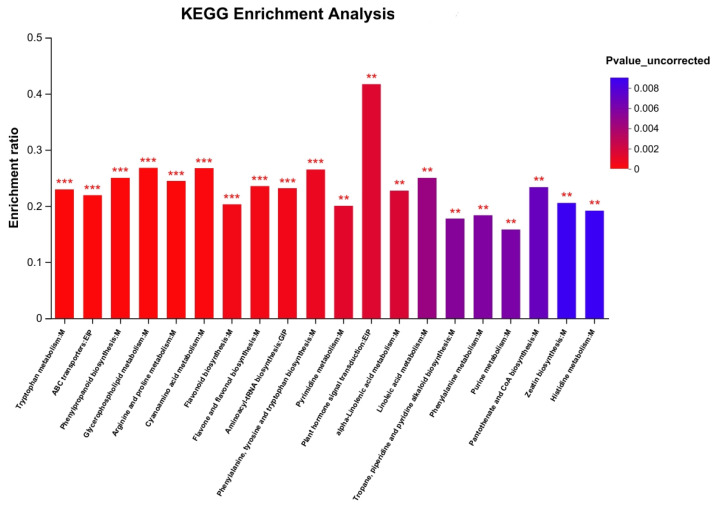
Enrichment analysis of metabolite KEGG in *Ziziphi Spinosae Semen*. The asterisks indicate statistically significant differences: ** *p* < 0.01, *** *p* < 0.001.

**Figure 5 plants-14-02739-f005:**
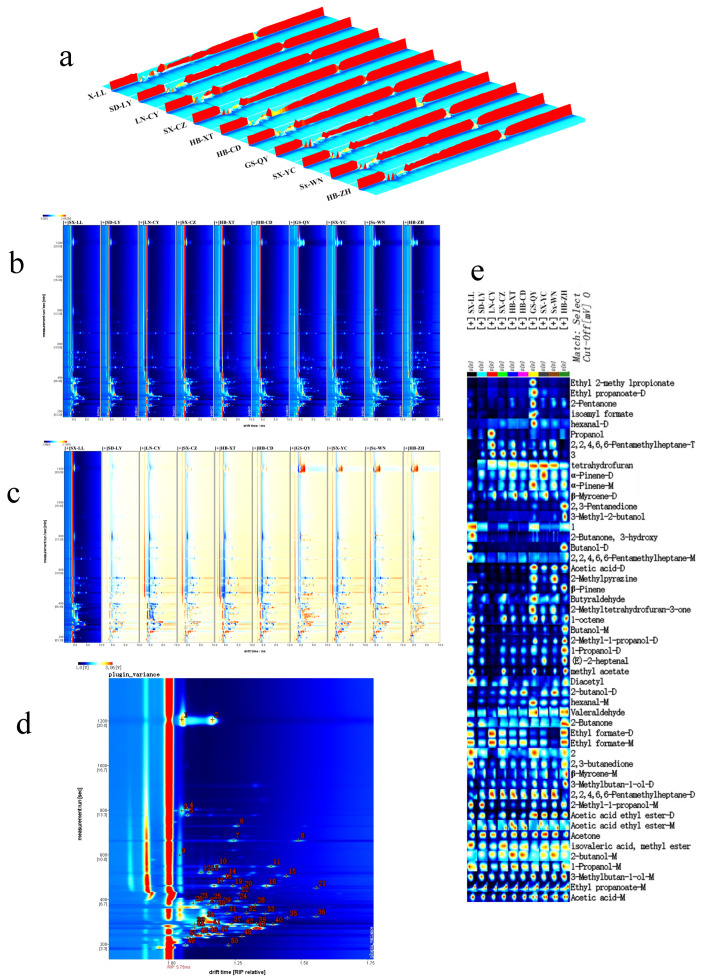
Geographic distribution of *Ziziphi Spinosae Semen* across production regions. (**a**) Three-dimensional topographic map; (**b**) topographic map; (**c**) sample differences comparison map. Plot based on retention time (s), relative ion travelling time (a.u.), and peak intensity (V) in the data. (**d**) Library search qualitative analysis plots of *Ziziphi Spinosae Semen* in different production areas. (**e**) Fingerprints of volatile compounds in *Ziziphi Spinosae Semen* from different production areas.

**Figure 6 plants-14-02739-f006:**
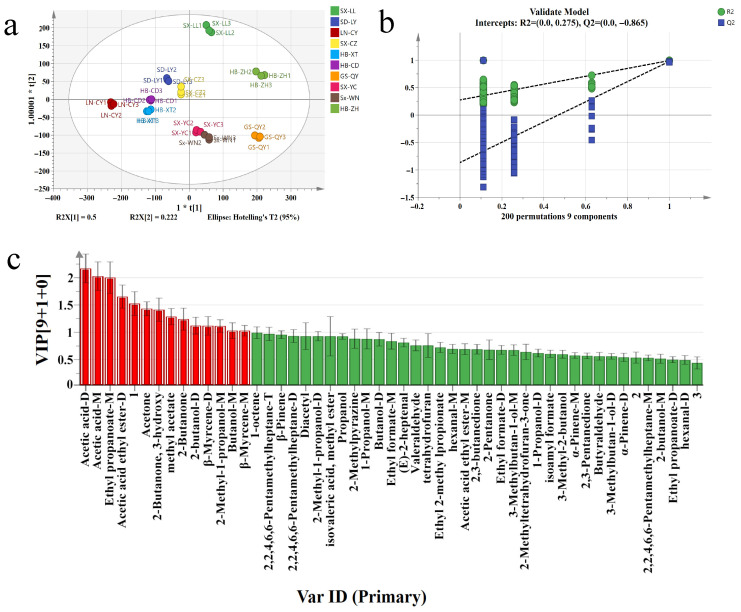
OPLS-DA analysis of *Ziziphi Spinosae Semen* across production regions using GC-IMS. (**a**) Plot of OPLS-DA scores; (**b**) plot of cross-validation of the 200-substitution test; (**c**) the red section represents the key differentiated compounds with VIP > 1.

**Figure 7 plants-14-02739-f007:**
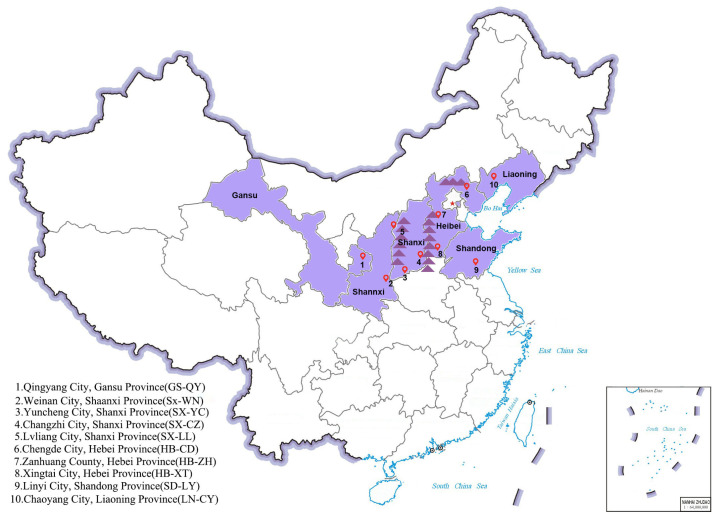
Geographical distribution of main production areas of *Ziziphi Spinosae Semen*. (The map is marked after downloading from the National Geographic Information Public Service Platform).

## Data Availability

The original contributions presented in this study are included in the article/Appendix A. Further inquiries can be directed to the corresponding authors.

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
