# Peer review of "Geospatial Metabolomics Unravel Regional Disparities in Sedative Compounds and Volatile Profiles of *Ziziphi Spinosae Semen* Across Chinese Production Areas"

_plants, 2025, doi:10.3390/plants14172739_

Round 1

Reviewer 1 Report

Comments and Suggestions for Authors

This manuscript presents a metabolomic comparison of Ziziphi Spinosae Semen (ZSS, sour jujube seeds), a widely used traditional Chinese medicine for mind soothing and insomnia relief. Samples were collected from 10 different production locations in China. The study makes a commendable effort to comprehensively evaluate the metabolite profiles of ZSS using both LC–MS and GC–IMS techniques, with reference to the key marker compounds jujuboside A, jujuboside B, and spinosyn. A total of 2,994 metabolites were characterized, and variations in both composition and quality were observed among the different locations.

Overall, the study is well designed and executed. The results are of good reference value for quality control of ZSS, and they may also provide guidance for future breeding strategies and research into the metabolic regulation of this medicinal plant.

The reviewer has the following comments and suggestions for improvement:

  1. Sample collection and preparation:
    The description of sample collection and replication needs to be clarified in greater detail. ZSS is mainly produced in northern China, and while samples were analyzed from 10 locations, the manuscript lacks sufficient information on:

    • How samples were collected (e.g., from wild plants or cultivated farms).

    • Whether samples were collected at the same or similar time.

    • Clarification of the statement “three replicates were obtained from three different sites in the same area.” Is “area” equivalent to “location”?

    • Whether samples from the “three different sites” were analyzed separately or pooled into one composite sample.

    • How the “three replicates” were treated before analysis—individually analyzed or pooled.

    • Most importantly, how many independent biological samples were analyzed per area/location.

  2. Figure legends:
    Figure legends should be more self-explanatory. For example, the legend of Figure 3 should specify how the data were processed (e.g., ANOVA, Tukey’s HSD test) and provide the number of samples per location (n = ?). The statistical methods and software used should also be clearly described in the Materials and Methods section.

  3. Figure readability:
    The text in the legend of Figure 3 is too small and difficult to read. Please enlarge for clarity.

  4. Discussion on representativeness:
    A discussion is needed regarding how representative the findings are, given the relatively small number of samples analyzed and the fact that some provinces were represented by only a single location.

Author Response

Comments 1:

  • Sample collection and preparation:
    The description of sample collection and replication needs to be clarified in greater detail. ZSS is mainly produced in northern China, and while samples were analyzed from 10 locations, the manuscript lacks sufficient information on:

    • How samples were collected (e.g., from wild plants or cultivated farms).

    • Whether samples were collected at the same or similar time.

    • Clarification of the statement “three replicates were obtained from three different sites in the same area.” Is “area” equivalent to “location”?

    • Whether samples from the “three different sites” were analyzed separately or pooled into one composite sample.

    • How the “three replicates” were treated before analysis—individually analyzed or pooled.

    • Most importantly, how many independent biological samples were analyzed per area/location.

Response 1: Thank you very much for pointing this out. Accordingly, we have revised Section 4.1 (Experimental Materials) in the Materials and Methods part of the manuscript (located on Page 11, Lines 318–328) to address the information that was previously lacking:

"Ziziphi Spinosae Semen (ZSS) samples were collected from wild jujube (Ziziphus acidojujuba Cheng et Liu–Z. jujuba Mill. var. spinosa Hu) plants across ten geographical locations in six major production provinces in China during the peak maturity period (September to October 2022). Detailed geographical coordinates and climatic characteristics of each sampling location are provided in Supplementary Table S1.

At each location, three biological replicates were collected within a proximate area. Each replicate consisted of seeds pooled from ten individual trees, yielding a total of 30 independent samples (10 locations × 3 replicates). Mature fruits were harvested, and seeds were carefully extracted after removal of the pulp and cracking of the stones. All samples were uniformly dried at 35 °C for 48 hours, sealed in airtight bags, and stored at −20 °C until further analysis to preserve metabolic integrity."

Additionally, descriptions of the climatic characteristics for each sampling site have been included in Supplementary Table S1.

Ten batches of sour jujube fruits collected from different geographical origin in China.

Ziziphi Spinosae Semen (ZSS)

Sample

Source

region

Longitude–latitude

Climate characteristics

GS-QY

Wildness

Qingyang, Gansu

35°42′N-107°38′E

Temperate continental climate; arid with low rainfall; abundant sunshine; mean annual temperature 9-11°C; annual precipitation 380-600 mm.

Sx-WN

Wildness

Weinan, Shaanxi

34°30′N-109°30′E

Warm temperate semi-humid monsoon climate; four distinct seasons; sufficient sunlight; mean annual temperature 12-14°C; annual precipitation ~600 mm.

SX-YC

Wildness

Yuncheng, Shanxi

35°20′N-110°59′E

Warm temperate continental monsoon climate; simultaneous rain and heat; mean annual temperature 12-13°C; annual precipitation 500-600 mm.

SX-CZ

Wildness

Changzhi, Shanxi

37°39′N-111°80′E

Warm temperate continental monsoon climate; mild winter and cool summer; large diurnal temperature variation; mean annual temperature 9-10°C; annual precipitation 550-650 mm.

SX-LL

Wildness

Lvliang, Shanxi

37°31′N-114°23′E

Temperate continental monsoon climate; cold winter and cool summer; mean annual temperature 7-9°C; annual precipitation 450-550 mm.

HB-CD

Wildness

Chengde, Hebei

42°10′N-118°42′E

Temperate continental monsoon climate; long winter and short summer; large diurnal temperature variation; mean annual temperature 7-9°C; annual precipitation 500-600 mm.

HB-ZH

Wildness

Zanhuang, Hebei

37°39′N-114°23′E

Warm temperate continental monsoon climate; dry and windy spring; hot and rainy summer; mean annual temperature 12-13°C; annual precipitation 500-600 mm.

HB-XT

Wildness

Xingtai, Hebei

37°40′N-114°30′E

Warm temperate continental monsoon climate; simultaneous rain and heat; mean annual temperature 13-14°C; annual precipitation 500-600 mm; concentrated summer rainfall.

SD-LY

Wildness

Linyi, Shandong

35°03′N-118°20′E

Warm temperate monsoon climate; simultaneous rain and heat; four distinct seasons; mean annual temperature 12.5-13.5°C; annual precipitation 800-900 mm.

LN-CY

Wildness

Chaoyang, Liaoning

42°01′N-121°39′E

Temperate continental monsoon climate; windy spring and autumn; cold winter and hot summer; annual temperature 8-9°C; annual precipitation 450-550 mm.

Comments 2:

  • Figure legends:
    Figure legends should be more self-explanatory. For example, the legend of Figure 3 should specify how the data were processed (e.g., ANOVA, Tukey’s HSD test) and provide the number of samples per location (n = ?). The statistical methods and software used should also be clearly described in the Materials and Methods section.

Response 2: hank you very much for pointing this out. We fully agree with this comment. Accordingly, we have made the following revisions:

  1. Added the following statement to the caption of Figure 3 (located on Page 5, Lines 143–144):
    "Values represent the average, and error bars represent standard errors. Different letters above the columns indicate significant differences (P < 0.05)."

  2. The following sentence has been added to Section 4 (Data Analysis) on Page 13, between Lines 420–423:

    ”Excel 2016 (Microsoft Corporation, USA) was employed for data organization and preliminary processing of the content of major medicinal components. A one-way analysis of variance (ANOVA) was carried out using SPSS 26.0 (SPSS Inc., Chicago, IL, USA), with significant differences among groups further compared by Duncan’s test (p < 0.05).“

Comments 3:

  • Figure readability:
    The text in the legend of Figure 3 is too small and difficult to read. Please enlarge for clarity.

Response 3: I fully agree with this point. Accordingly, we have modified Figure 3 on Page 5 (between Lines 141–142) by enlarging the figure to address its legibility issues.

Comments 4:

  • Discussion on representativeness:
    A discussion is needed regarding how representative the findings are, given the relatively small number of samples analyzed and the fact that some provinces were represented by only a single location.

Response 4: We fully agree with this comment. Accordingly, we have added a paragraph in the Discussion section (Section 3) of the manuscript, located on Pages 10–11 (Lines 305–315), to address the issue of the relatively small sample size and the fact that some provinces were represented by only a single sampling site.

“Although the samples covered 10 production regions across six provinces (with 30 biological replicates in total), meeting the basic sample size requirements for omics studies, certain provinces (e.g., GS-QY, SD-LY, and LN-CY) contained only a single sampling site, which limited comprehensive characterization of intra-province chemical diversity. Furthermore, the effects of spatial heterogeneity of wild resources and interannual climate fluctuations on metabolite profiles could not be fully captured due to the single-year sampling design. Future studies should expand the sampling to include more sites across broader geographical ranges, incorporate multi-year dynamic sampling, and integrate genetic analytical approaches such as DNA barcoding or population genomics to systematically clarify the interactive effects of genetic and environmental factors on the metabolic diversity of ZSS.“

Reviewer 2 Report

Comments and Suggestions for Authors

Thank you for submitting the manuscript to the journal. This is a really excellent piece of work. The methodology is robust, the data is comprehensive, and your findings make a significant contribution to the field. The core discovery that climate directly shapes the metabolic profile of Ziziphi Spinosae Semen is both profound and highly valuable.

Here are a few comments and suggestions:

  1. You've clearly shown that climate dictates the metabolic profiles. To make the story even stronger, could you speculate a bit more on how specific weather conditions (like the temperature or rainfall) directly influence the key pathways you identified (like tryptophan metabolism)? Even some informed speculation would help build a more compelling mechanistic narrative for the reader.
  2. You mention that genetic differences between populations might be a factor, which is a fair point. Since you don't present genetic data, it would strengthen the paper to briefly acknowledge that it's currently hard to disentangle the environmental and genetic influences. A sentence or two suggesting how future studies could explore this would be a perfect addition.
  3. In the abstract, you group terpenoids with lipids and amino acids as "primary nutrients." For clarity, it might be helpful to rephrase this. Lipids and amino acids are classic nutrients, while terpenoids are more often seen as secondary metabolites with medicinal properties. A slight tweak here will ensure your categorization is crystal clear for everyone.
  4. The figures are central to your story. Please double-check that all of them (especially the key ones like Figures 2 and 5) are in the highest possible resolution with super-clear labels. Likewise, the writing is already very good, just a careful proofread for the tiniest of grammatical tweaks and stylistic consistency will give it that final polish.

I have decided to accept after a chance to address a few minor points.

Author Response

Comments 1:

You've clearly shown that climate dictates the metabolic profiles. To make the story even stronger, could you speculate a bit more on how specific weather conditions (like the temperature or rainfall) directly influence the key pathways you identified (like tryptophan metabolism)? Even some informed speculation would help build a more compelling mechanistic narrative for the reader.

Response 1:I fully agree with this point. Accordingly, we have added a paragraph in Section 2.4 (Metabolic Pathway Enrichment) of the manuscript, located on Page 6 (Lines 151–156), to further speculate on the potential influence of climatic factors on tryptophan metabolism, thereby strengthening the narrative:

"The tryptophan biosynthetic pathway plays a critical role in coordinating plant growth and stress responses.[14]. As the wild jujube tree primarily grows in the mountainous regions of northern China, adverse conditions such as low temperatures and significant diurnal temperature variations may promote tryptophan accumulation, thereby enhancing the synthesis of downstream neuroactive metabolites[15]."

Additionally, descriptions of the climatic characteristics for each sampling site have been included in Supplementary Table S1.

Ten batches of sour jujube fruits collected from different geographical origin in China.

Ziziphi Spinosae Semen (ZSS)

Sample

Source

region

Longitude–latitude

Climate characteristics

GS-QY

Wildness

Qingyang, Gansu

35°42′N-107°38′E

Temperate continental climate; arid with low rainfall; abundant sunshine; mean annual temperature 9-11°C; annual precipitation 380-600 mm.

Sx-WN

Wildness

Weinan, Shaanxi

34°30′N-109°30′E

Warm temperate semi-humid monsoon climate; four distinct seasons; sufficient sunlight; mean annual temperature 12-14°C; annual precipitation ~600 mm.

SX-YC

Wildness

Yuncheng, Shanxi

35°20′N-110°59′E

Warm temperate continental monsoon climate; simultaneous rain and heat; mean annual temperature 12-13°C; annual precipitation 500-600 mm.

SX-CZ

Wildness

Changzhi, Shanxi

37°39′N-111°80′E

Warm temperate continental monsoon climate; mild winter and cool summer; large diurnal temperature variation; mean annual temperature 9-10°C; annual precipitation 550-650 mm.

SX-LL

Wildness

Lvliang, Shanxi

37°31′N-114°23′E

Temperate continental monsoon climate; cold winter and cool summer; mean annual temperature 7-9°C; annual precipitation 450-550 mm.

HB-CD

Wildness

Chengde, Hebei

42°10′N-118°42′E

Temperate continental monsoon climate; long winter and short summer; large diurnal temperature variation; mean annual temperature 7-9°C; annual precipitation 500-600 mm.

HB-ZH

Wildness

Zanhuang, Hebei

37°39′N-114°23′E

Warm temperate continental monsoon climate; dry and windy spring; hot and rainy summer; mean annual temperature 12-13°C; annual precipitation 500-600 mm.

HB-XT

Wildness

Xingtai, Hebei

37°40′N-114°30′E

Warm temperate continental monsoon climate; simultaneous rain and heat; mean annual temperature 13-14°C; annual precipitation 500-600 mm; concentrated summer rainfall.

SD-LY

Wildness

Linyi, Shandong

35°03′N-118°20′E

Warm temperate monsoon climate; simultaneous rain and heat; four distinct seasons; mean annual temperature 12.5-13.5°C; annual precipitation 800-900 mm.

LN-CY

Wildness

Chaoyang, Liaoning

42°01′N-121°39′E

Temperate continental monsoon climate; windy spring and autumn; cold winter and hot summer; annual temperature 8-9°C; annual precipitation 450-550 mm.

Comments 2:

You mention that genetic differences between populations might be a factor, which is a fair point. Since you don't present genetic data, it would strengthen the paper to briefly acknowledge that it's currently hard to disentangle the environmental and genetic influences. A sentence or two suggesting how future studies could explore this would be a perfect addition.

Response 2:We fully agree with this comment and sincerely appreciate your suggestion. Accordingly, we have added a paragraph in the Discussion section (Section 3) of the manuscript, specifically on Pages 10–11 (Lines 305–315), to provide further clarification on this issue:

"Although the samples covered 10 production regions across six provinces (with 30 biological replicates in total), meeting the basic sample size requirements for omics studies, certain provinces (e.g., GS-QY, SD-LY, and LN-CY) contained only a single sampling site, which limited comprehensive characterization of intra-province chemical diversity. Furthermore, the effects of spatial heterogeneity of wild resources and interannual climate fluctuations on metabolite profiles could not be fully captured due to the single-year sampling design. Future studies should expand the sampling to include more sites across broader geographical ranges, incorporate multi-year dynamic sampling, and integrate genetic analytical approaches such as DNA barcoding or population genomics to systematically clarify the interactive effects of genetic and environmental factors on the metabolic diversity of ZSS."

Comments 3:

In the abstract, you group terpenoids with lipids and amino acids as "primary nutrients." For clarity, it might be helpful to rephrase this. Lipids and amino acids are classic nutrients, while terpenoids are more often seen as secondary metabolites with medicinal properties. A slight tweak here will ensure your categorization is crystal clear for everyone.

Response 3:I fully agree with this comment. Accordingly, we have revised the Abstract section of the manuscript (located on Page 1, Lines 24–25) to clearly distinguish lipids and amino acids as classical nutrients from terpenoids, which are categorized as secondary metabolites, thereby improving the clarity of the classification:

"Lipids and amino acids were the primary nutrients, while terpenoids were the most abundant class of secondary metabolites."

Comments 4:

The figures are central to your story. Please double-check that all of them (especially the key ones like Figures 2 and 5) are in the highest possible resolution with super-clear labels. Likewise, the writing is already very good, just a careful proofread for the tiniest of grammatical tweaks and stylistic consistency will give it that final polish.

Response 4:Thank you very much for pointing this out. Accordingly, we have enlarged both Figure 2 (located on Page 4, Lines 121–122) and Figure 5 (located on Page 8, Lines 227–228) to improve their clarity.
